# LEARNING TO SHARE IN MULTI-AGENT REINFORCEMENT LEARNING

## ABSTRACT

In this paper, we study the problem of networked multi-agent reinforcement learning (MARL), where a number of agents are deployed as a partially connected network. Networked MARL requires all agents make decision in a decentralized manner to optimize a global objective with restricted communication between neighbors over the network. We propose a hierarchically decentralized MARL method, *LToS*, which enables agents to learn to dynamically share reward with neighbors so as to encourage agents to cooperate on the global objective. For each agent, the high-level policy learns how to share reward with neighbors to decompose the global objective, while the low-level policy learns to optimize local objective induced by the high-level policies in the neighborhood. The two policies form a bi-level optimization and learn alternately. We empirically demonstrate that LToS outperforms existing methods in both social dilemma and two networked MARL scenarios.

## 1 INTRODUCTION

In multi-agent reinforcement learning (MARL), there are multiple agents interacting with the environment via their joint action to cooperatively optimize an objective. Many methods of centralized training and decentralized execution (CTDE) have been proposed for cooperative MARL, such as VDN (Sunehag et al., 2018), QMIX (Rashid et al., 2018), and QTRAN (Son et al., 2019). However, these methods suffer from the overgeneralization issue (Palmer et al., 2018; Castellini et al., 2019). Moreover, they may not easily scale up with the number of agents due to centralized learning (Qu et al., 2020a).

In many MARL applications, there are a large number of agents that are deployed as a partially connected network and collaboratively make decisions to optimize the globally averaged return, such as smart grids (Dall'Anese et al., 2013), network routing (Jiang et al., 2020), traffic signal control (Chu et al., 2020), and IoT (Xu et al., 2019). To deal with such scenarios, networked MARL is formulated to decompose the dependency among all agents into dependencies between only neighbors in such scenarios. To avoid decision-making with insufficient information, agents are permitted to exchange messages with neighbors over the network. In such settings, it is feasible for agents to learn to make decisions in a decentralized way (Zhang et al., 2018; Qu et al., 2020b). However, there are still difficulties of dependency if anyone attempts to make decision independently, e.g., *prisoner's dilemma* and *tragedy of the commons* (Pérolat et al., 2017). Existing methods tackle these problems by consensus update of value function (Zhang et al., 2018), credit assignment (Wang et al., 2020), or reward shaping (Chu et al., 2020). However, these methods rely on either access to global state and joint action (Zhang et al., 2018) or handcrafted reward functions (Wang et al., 2020; Chu et al., 2020).

Inspired by the fact that *sharing* plays a key role in human's learning of cooperation, in this paper, we propose *Learning To Share* (LToS), a hierarchically decentralized learning method for networked MARL. LToS enables agents to learn to dynamically share reward with neighbors so as to collaboratively optimize the global objective. The high-level policies decompose the global objective into local ones by determining how to share their rewards, while the low-level policies optimize local objectives induced by the high-level policies. LToS learns in a decentralized manner, and we prove that the high-level policies are a mean-field approximation of the joint high-level policy. Moreover, the high-level and low-level policies form a bi-level optimization and alternately learn to optimize

the global objective. LToS is easy to implement and currently realized by DDPG (Lillicrap et al., 2016) as the high-level policy and DGN (Jiang et al., 2020) as the low-level policy. We empirically demonstrate that LToS outperforms existing methods for networked MARL in both social dilemma and two real-world scenarios. To the best of our knowledge, LToS is the first to learn to share reward for global optimization in networked MARL.

## 2 RELATED WORK

There are many recent studies for collaborative MARL. Most of them adopt centralized training and decentralized execution, such as COMA (Foerster et al., 2018), VDN (Sunehag et al., 2018), QMIX (Rashid et al., 2018), and QTRAN (Son et al., 2019). Many are constructed on the basis of factorizing the joint Q-function by assuming additivity (Sunehag et al., 2018), monotonicity (Rashid et al., 2018), or factorizable tasks (Son et al., 2019). However, they are learned in a centralized way and hence may not easily scale up with the number of agents in networked MARL (Qu et al., 2020a). Moreover, these factorized methods suffer from the overgeneralization issue (Palmer et al., 2018; Castellini et al., 2019).

Other studies focus on decentralized training specifically in networked MARL, to which our work is more closely related. Zhang et al. (2018) proposed consensus update of value function, but it requires global state at each agent, which is usually unavailable in decentralized training. Chu et al. (2020) introduced a spatial discount factor to capture the influence between agents, but the spatial discount factor remains hand-tuned. Sodomka et al. (2013) and Peysakhovich & Lerer (2018b) involved the concept of transferable utility to encourage cooperation, and Peysakhovich & Lerer (2018a) resorted to game theory and gave more complex reward designs. However, these methods cannot be extended beyond two-player games. Hughes et al. (2018) proposed the inequity aversion model to balance agents' selfish desire and social fairness. Wang et al. (2020) considered to learn the Shapley value as the credit assignment. However, these methods still rely on hand-crafted reward designs. Mguni et al. (2019) added an extra part to the original reward as non-potential based reward shaping and used Bayesian optimization to induce the convergence to a desirable equilibrium between agents. However, the extra part remains fixed during an episode, which makes it less capable of dealing with dynamic environments. Moreover, the reward shaping alters the original optimization problem.

## 3 BACKGROUND

### 3.1 NETWORKED MULTI-AGENT REINFORCEMENT LEARNING

Assume $N$ agents interact with an environment. Let $\mathcal{V} = \{1, 2, \cdots, N\}$ be the set of agents. The multi-agent system is modeled as an undirected graph $\mathcal{G}(\mathcal{V}, \mathcal{E})$, where each agent $i$ serves as vertex $i$ and $\mathcal{E} \subseteq \mathcal{V} \times \mathcal{V}$ is the set of all edges. Two agents $i, j \in \mathcal{V}$ can communicate with each other if and only if $e_{ij} = (i, j) \in \mathcal{E}$. We denote *agent $i$ and its all neighbors in the graph together as a set* $\mathcal{N}_i$. The state of the environment $s \in \mathcal{S}$ transitions upon joint action $\boldsymbol{a} \in \mathcal{A}$ according to transition probability $\mathcal{P}_a : \mathcal{S} \times \mathcal{A} \times \mathcal{S} \rightarrow [0, 1]$, where joint action set $\mathcal{A} = \times_{i \in \mathcal{V}} \mathcal{A}_i$. Each agent $i$ has a policy $\pi_i \in \Pi_i : \mathcal{S} \times \mathcal{A}_i \rightarrow [0, 1]$, and we denote the joint policy of all agents as $\boldsymbol{\pi} \in \Pi = \times_{i \in \mathcal{V}} \Pi_i$. For networked MARL, a common and realistic assumption is that the reward of each agent $i$ just depends on its action and the actions of its neighbors (Qu et al., 2020a), *i.e.*, $r_i(s, \boldsymbol{a}) = r_i(s, a_{\mathcal{N}_i})$. Moreover, each agent $i$ may only obtain partial observation $o_i \in \mathcal{O}_i$, but can approximate the state by the observations of $\mathcal{N}_i$ (Jiang et al., 2020) or the observation history (Chu et al., 2020), which are all denoted by $o_i$ for simplicity. The *global objective* is to maximize the sum of cumulative rewards of all agents , *i.e.*, $\sum_{t=0}^{\infty} \sum_{i=1}^{N} \gamma^t r_i^t$.

### 3.2 MARKOV GAME

In such a setting, each agent can individually maximizes its own expected return, which is known as Markov game. This may lead to stable outcome or Nash equilibrium, which however is usually sub-optimal. Given $\boldsymbol{\pi}$, the value function of agent $i$ is given by

$$v_i^{\boldsymbol{\pi}}(s) = \sum_{\boldsymbol{a}} \boldsymbol{\pi}(\boldsymbol{a}|s) \sum_{s'} p_a(s'|s, \boldsymbol{a})[r_i + \gamma v_i^{\boldsymbol{\pi}}(s')], \tag{1}$$

where $p_a \in \mathcal{P}_a$ describes the state transitions. A Nash equilibrium is defined as (Mguni et al., 2019)

$$v_i^{(\pi_i, \pi_{-i})}(s) \geq v_i^{(\pi_i', \pi_{-i})}(s), \quad \forall \pi_i' \in \Pi_i, \forall s \in \mathcal{S}, \forall i \in \mathcal{V}, \tag{2}$$

where $\pi_{-i} = \times_{j \in \mathcal{V} \setminus \{i\}} \pi_j$.

## 4 METHOD

The basic idea of LToS is to enable agents to learn how to share reward with neighbors such that agents are encouraged to collaboratively optimize the global objective in networked MARL. LToS is a decentralized hierarchy. At each agent, the high-level policy determines the weights of reward sharing based on low-level policies while the low-level policy directly interacts with the environment to optimize local objective induced by high-level policies. Therefore, they form a bi-level optimization and alternately learn towards the global objective.

### 4.1 REWARD SHARING

The intuition of reward sharing is that if agents share their rewards with others, each agent has to consider the consequence of its actions on others, and thus it promotes cooperation. In networked MARL, as the reward of an agent is assumed to depend on the actions of neighbors, we allow reward sharing between neighboring agents.

For the graph of $\mathcal{V}$, we additionally define a set of directed edges, $\mathcal{D}$, constructed from $\mathcal{E}$. Specifically, we add a loop $d_{ii} \in \mathcal{D}$ for each agent $i$ and split each undirected edge $e_{ij} \in \mathcal{E}$ into two directed edges: $d_{ij} = (i, j)$ and $d_{ji} = (j, i) \in \mathcal{D}$. Each agent $i$ determines a weight $w_{ij} \in [0, 1]$ for each directed edge $d_{ij}, \forall j \in \mathcal{N}_i$, subject to the constraint $\sum_{j \in \mathcal{N}_i} w_{ij} = 1$, so that $w_{ij}$ proportion of agent $i$'s environment reward $r_i$ will be shared to agent $j$. Let $\boldsymbol{w} \in \mathcal{W} = \times_{d_{ij} \in \mathcal{D}} w_{ij}$ be the weights of the graph. Therefore, the shaped reward after sharing for each agent $i$ is defined as

$$r_i^{\boldsymbol{w}} = \sum_{j \in \mathcal{N}_i} w_{ji} r_j. \tag{3}$$

### 4.2 HIERARCHY

Assume there is a joint high-level policy $\phi \in \Phi : \mathcal{S} \times \mathcal{W} \to [0, 1]$ to determine $\boldsymbol{w}$. Given $\phi$ and $\boldsymbol{w}$, we can define the value function of $\boldsymbol{\pi}$ at each agent $i$ based on (1) as

$$v_i^{\boldsymbol{\pi}}(s; \phi) = \sum_{\boldsymbol{w}} \phi(\boldsymbol{w}|s) \sum_{\boldsymbol{a}} \boldsymbol{\pi}(\boldsymbol{a}|s, \boldsymbol{w}) \sum_{s'} p_a(s'|s, \boldsymbol{a})[r_i^{\boldsymbol{w}} + \gamma v_i^{\boldsymbol{\pi}}(s'; \phi)], \tag{4}$$

$$v_i^{\boldsymbol{\pi}}(s; \boldsymbol{w}, \phi) = \sum_{\boldsymbol{a}} \boldsymbol{\pi}(\boldsymbol{a}|s, \boldsymbol{w}) \sum_{s'} p_a(s'|s, \boldsymbol{a})[r_i^{\boldsymbol{w}} + \gamma v_i^{\boldsymbol{\pi}}(s'; \phi)]. \tag{5}$$

We express $\boldsymbol{w}$ as a discrete action for simplicity. It also holds for continuous action as long as we change all the summations to integrals.

Let $V_{\mathcal{V}}^{\phi}(s; \boldsymbol{\pi}) \doteq \sum_{i \in \mathcal{V}} v_i^{\boldsymbol{\pi}}(s; \phi)$ and $Q_{\mathcal{V}}^{\phi}(s, \boldsymbol{w}; \boldsymbol{\pi}) \doteq \sum_{i \in \mathcal{V}} v_i^{\boldsymbol{\pi}}(s; \boldsymbol{w}, \phi)$.

**Proposition 4.1.** *Given $\boldsymbol{\pi}$, $V_{\mathcal{V}}^{\phi}(s; \boldsymbol{\pi})$ and $Q_{\mathcal{V}}^{\phi}(s, \boldsymbol{w}; \boldsymbol{\pi})$ are respectively the value function and action-value function of $\phi$.*

*Proof.* Let $r_i^{\phi} \doteq \sum_{\boldsymbol{a}} \boldsymbol{\pi}(\boldsymbol{a}|s, \boldsymbol{w}) r_i^{\boldsymbol{w}}$ and $p_w(s'|s, \boldsymbol{w}) \doteq \sum_{\boldsymbol{a}} \boldsymbol{\pi}(\boldsymbol{a}|s, \boldsymbol{w}) p_a(s'|s, \boldsymbol{a})$. As commonly assumed the reward is deterministic given $s$ and $\boldsymbol{a}$, from (4), we have,

$$v_i^{\boldsymbol{\pi}}(s; \phi) = \sum_{\boldsymbol{w}} \phi(\boldsymbol{w}|s) \sum_{\boldsymbol{a}} \boldsymbol{\pi}(\boldsymbol{a}|s, \boldsymbol{w})[r_i^{\boldsymbol{w}} + \sum_{s'} p_a(s'|s, \boldsymbol{a}) \gamma v_i^{\boldsymbol{\pi}}(s'; \phi)] \tag{6}$$

$$= \sum_{\boldsymbol{w}} \phi(\boldsymbol{w}|s) \sum_{s'} p_w(s'|s, \boldsymbol{w})[r_i^{\phi} + \gamma v_i^{\boldsymbol{\pi}}(s'; \phi)], \tag{7}$$

where $p_w \in \mathcal{P}_w : \mathcal{S} \times \mathcal{W} \times \mathcal{S} \to [0, 1]$ describes the state transitions given $\boldsymbol{\pi}$.

Let $r_{\mathcal{V}}^{\phi} \doteq \sum_{i \in \mathcal{V}} r_i^{\phi}$, and from (7) we have

$$V_{\mathcal{V}}^{\phi}(s; \boldsymbol{\pi}) = \sum_{i \in \mathcal{V}} \sum_{\boldsymbol{w}} \phi(\boldsymbol{w}|s) \sum_{s'} p_w(s'|s, \boldsymbol{w})[r_i^{\phi} + \gamma v_i^{\boldsymbol{\pi}}(s'; \phi)] \tag{8}$$

$$= \sum_{\boldsymbol{w}} \phi(\boldsymbol{w}|s) \sum_{s'} p_w(s'|s, \boldsymbol{w})[\sum_{i \in \mathcal{V}} r_i^{\phi} + \gamma \sum_{i \in \mathcal{V}} v_i^{\boldsymbol{\pi}}(s'; \phi)] \tag{9}$$

$$= \sum_{\boldsymbol{w}} \phi(\boldsymbol{w}|s) \sum_{s'} p_w(s'|s, \boldsymbol{w})[r_{\mathcal{V}}^{\phi} + \gamma V_{\mathcal{V}}^{\phi}(s'; \boldsymbol{\pi})], \tag{10}$$

and similarly,

$$Q_{\mathcal{V}}^{\phi}(s, \boldsymbol{w}; \boldsymbol{\pi}) = \sum_{i \in \mathcal{V}} \sum_{s'} p_w(s'|s, \boldsymbol{w})[r_i^{\phi} + \gamma \sum_{\boldsymbol{w}'} \phi(\boldsymbol{w}'|s') v_i^{\boldsymbol{\pi}}(s'; \boldsymbol{w}', \phi)] \tag{11}$$

$$= \sum_{s'} p_w(s'|s, \boldsymbol{w})[\sum_{i \in \mathcal{V}} r_i^{\phi} + \gamma \sum_{\boldsymbol{w}'} \phi(\boldsymbol{w}'|s') \sum_{i \in \mathcal{V}} v_i^{\boldsymbol{\pi}}(s'; \boldsymbol{w}', \phi)] \tag{12}$$

$$= \sum_{s'} p_w(s'|s, \boldsymbol{w})[r_{\mathcal{V}}^{\phi} + \gamma \sum_{\boldsymbol{w}'} \phi(\boldsymbol{w}'|s') Q_{\mathcal{V}}^{\phi}(s', \boldsymbol{w}'; \boldsymbol{\pi})]. \tag{13}$$

Moreover, from the definitions of $r_i^{\boldsymbol{w}}$ and $r_i^{\phi}$ we have

$$r_{\mathcal{V}}^{\phi} = \sum_{\boldsymbol{a}} \boldsymbol{\pi}(\boldsymbol{a}|s, \boldsymbol{w}) \sum_{i \in \mathcal{V}} r_i^{\boldsymbol{w}} = \sum_{\boldsymbol{a}} \boldsymbol{\pi}(\boldsymbol{a}|s, \boldsymbol{w}) \sum_{i \in \mathcal{V}} \sum_{j \in \mathcal{N}_i} w_{ji} r_j \tag{14}$$

$$= \sum_{\boldsymbol{a}} \boldsymbol{\pi}(\boldsymbol{a}|s, \boldsymbol{w}) \sum_{(i,j) \in \mathcal{D}} w_{ij} r_i = \sum_{\boldsymbol{a}} \boldsymbol{\pi}(\boldsymbol{a}|s, \boldsymbol{w}) \sum_{i \in \mathcal{V}} r_i, \tag{15}$$

Thus, given $\boldsymbol{\pi}$, $V_{\mathcal{V}}^{\phi}(s)$ and $Q_{\mathcal{V}}^{\phi}(s, \boldsymbol{w})$ are respectively the value function and action-value function of $\phi$ in terms of the sum of expected cumulative rewards of all agents, *i.e.*, the global objective. $\square$

Proposition 4.1 implies that $\phi$ directly optimizes the global objective by generating $\boldsymbol{w}$ given $\boldsymbol{\pi}$. Unlike existing hierarchical RL methods, we can directly construct the value function and action-value function of $\phi$ based on the value function of $\boldsymbol{\pi}$ at each agent.

As $\phi$ optimizes the global objective given $\boldsymbol{\pi}$ while $\pi_i$ optimizes the shaped reward individually at each agent given $\phi$ (assuming $\boldsymbol{\pi}$ convergent to Nash equilibrium or stable outcome, denoted as lim), they form a bi-level optimization. Let $J_{\phi}(\boldsymbol{\pi})$ and $J_{\boldsymbol{\pi}}(\phi)$ denote the objectives of $\phi$ and $\boldsymbol{\pi}$ respectively. The bi-level optimization can be formulated as follows,

$$\begin{aligned} \max_{\phi} \quad & J_{\phi}(\boldsymbol{\pi}^*(\phi)) \\ s.t. \quad & \boldsymbol{\pi}^*(\phi) = \arg \lim_{\boldsymbol{\pi}} J_{\boldsymbol{\pi}}(\phi). \end{aligned} \tag{16}$$

### 4.3 Decentralized Learning

**Proposition 4.2.** *The joint high-level policy $\phi$ can be learned in a decentralized manner, and the decentralized high-level policies of all agents form a mean-field approximation of $\phi$.*

*Proof.* Let $d_{ij} \in \mathcal{D}$ serve as a vertex with action $w_{ij}$ and reward $w_{ij} r_i$ in a new graph $\mathcal{G}'$. Each vertex has its own local policy $\phi_{ij}(w_{ij}|s)$, and we can verify their independence by means of Markov Random Field. For $\forall i \in \mathcal{V}$, $\{d_{ij}|j \in \mathcal{N}_i\}$ should form a fully connected subgraph in $\mathcal{G}'$, because their actions are subject to the constraint $\sum_{j \in \mathcal{N}_i} w_{ij} = 1$. As $d_{ij} \in \mathcal{G}'$ only connects to $\{d_{ik}|k \in \mathcal{N}_i \setminus \{j\}\}$, the fully connected subgraph is also a maximal clique. According to Hammersley–Clifford theorem (Hammersley & Clifford, 1971), we have $\phi(\boldsymbol{w}|s) \approx \prod_{i \in \mathcal{V}} \phi_i(w_i^{\text{out}}|s)$, where $w_i^{\text{out}} = \{w_{ij}|j \in \mathcal{N}_i\}$. $\square$

Proposition 4.1 and 4.2 indicate that for each agent $i$, the low-level policy simply learns a local $\pi_i(a_i|s, w_i^{\text{in}})$, where $w_i^{\text{in}} = \{w_{ji}|j \in \mathcal{N}_i\}$, to optimize the cumulative reward of $r_i^{\boldsymbol{w}}$, since $r_i^{\boldsymbol{w}}$ is

fully determined by $w_i^{\text{in}}$ according to (3) and denoted as $r_i^w$ from now on. And the high-level policy $\phi_i$ just needs to locally determine $w_i^{\text{out}}$ to maximize the cumulative reward of $r_i^{\phi}$ simplified as $r_i^{\phi}$.

Therefore, for decentralized learning, (16) can be decomposed locally for each agent $i$ as

$$\begin{aligned}
\max_{\phi_i} \quad & J_{\phi_i}(\phi_{-i}, \pi_1^*(\boldsymbol{\phi}), \cdots, \pi_N^*(\boldsymbol{\phi})) \\
s.t. \quad & \pi_i^*(\boldsymbol{\phi}) = \arg\max_{\pi_i} J_{\pi_i}(\pi_{-i}, \phi_1(\boldsymbol{\pi}), \cdots, \phi_N(\boldsymbol{\pi})),
\end{aligned} \tag{17}$$

We abuse the notation and let $\phi$ and $\pi$ also denote their parameterizations respectively. To solve the optimization, we have

$$\begin{aligned}
\nabla_{\phi_i} & J_{\phi_i}(\phi_{-i}, \pi_1^*(\boldsymbol{\phi}), \cdots, \pi_N^*(\boldsymbol{\phi})) \\
& \approx \nabla_{\phi_i} J_{\phi_i}(\phi_{-i}, \pi_1 + \alpha \nabla_{\pi_1} J_{\pi_1}(\boldsymbol{\phi}), \cdots, \pi_N + \alpha \nabla_{\pi_N} J_{\pi_N}(\boldsymbol{\phi})),
\end{aligned} \tag{18}$$

where $\alpha$ is the learning rate for the low-level policy. Let $\pi_i'$ denote $\pi_i + \alpha \nabla_{\pi_i} J_{\pi_i}(\boldsymbol{\phi})$, we have

$$\begin{aligned}
\nabla_{\phi_i} & J_{\phi_i}(\phi_{-i}, \pi_1^*(\boldsymbol{\phi}), \cdots, \pi_N^*(\boldsymbol{\phi})) \\
& \approx \nabla_{\phi_i} J_{\phi_i}(\phi_{-i}, \pi_1', \cdots, \pi_N') + \alpha \sum_{j=1}^{N} \nabla_{\phi_i, \pi_j}^2 J_{\pi_j}(\boldsymbol{\phi}) \nabla_{\pi_j'} J_{\phi_i}(\phi_{-i}, \pi_1', \cdots, \pi_N').
\end{aligned} \tag{19}$$

The second-order derivative is neglected due to high computational complexity, without incurring significant performance drop such as in meta-learning (Finn et al., 2017) and neural architecture search (Liu et al., 2019). Similarly, we have

$$\begin{aligned}
\nabla_{\pi_i} & J_{\pi_i}(\pi_{-i}, \phi_1^*(\boldsymbol{\pi}), \cdots, \phi_N^*(\boldsymbol{\pi})) \\
& \approx \nabla_{\pi_i} J_{\pi_i}(\pi_{-i}, \phi_1 + \beta \nabla_{\phi_1} J_{\phi_1}(\boldsymbol{\pi}), \cdots, \phi_N + \beta \nabla_{\phi_N} J_{\phi_N}(\boldsymbol{\pi})),
\end{aligned} \tag{20}$$

where $\beta$ is the learning rate of the high-level policy. Therefore, we can solve the bi-level optimization (16) by the first-order approximations in a decentralized way. For each agent $i$, $\phi_i$ and $\pi_i$ are alternately updated.

In distributed learning, as each agent $i$ usually does not have access to state, we further approximate $\phi_i(w_i^{\text{out}}|s)$ and $\pi_i(a_i|s, w_i^{\text{in}})$ by $\phi_i(w_i^{\text{out}}|o_i)$ and $\pi_i(a_i|o_i, w_i^{\text{in}})$, respectively. Moreover, in network MARL as each agent $i$ is closely related to neighboring agents, (17) can be further seen as $\pi_i$ maximizes the cumulative discounted reward of $r_i^w$ given $\phi_{\mathcal{N}_i}$, where $\phi_{\mathcal{N}_i} = \times_{j \in \mathcal{N}_i} \phi_j$, and $\phi_i$ optimizes the cumulative discounted reward of $r_i^w$ given $\pi_{\mathcal{N}_i}$ (*i.e.*, $r_i^{\phi}$), where $\pi_{\mathcal{N}_i} = \times_{j \in \mathcal{N}_i} \pi_j$. During training, $\pi_{\mathcal{N}_i}$ and $\phi_{\mathcal{N}_i}$ are implicitly considered by interactions of $w_i^{\text{out}}$ and $w_i^{\text{in}}$ respectively. The architecture of LToS is illustrated in Figure 1. At each timestep, the high-level policy of each agent $i$ makes a decision of action $w_i^{\text{out}}$ as the weights of reward sharing based on the observation. Then, the low-level policy takes the observation and $w_i^{\text{in}}$

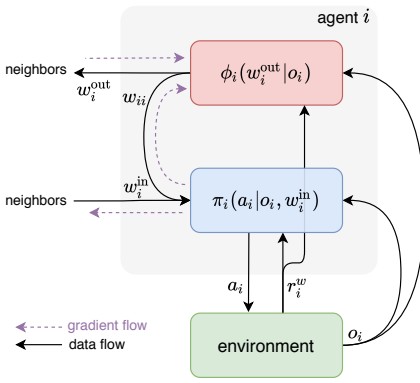

Figure 1: LToS

as an input and outputs the action. Agent $i$ obtains the shaped reward according to $w_i^{\text{in}}$ for both the high-level and low-level policies. The gradients are backpropagated along purple dotted lines.

Further, from Proposition 4.1, we have $q_i^{\phi_i}(s, w_i^{\text{out}}; \pi_{\mathcal{N}_i}) = v_i^{\pi_i}(s; w_i^{\text{in}}, \phi_{\mathcal{N}_i})$, where $q_i^{\phi_i}$ is the action-value function of $\phi_i$ given $\pi_{\mathcal{N}_i}$, $v_i^{\pi_i}$ is the value function of $\pi_i$ given $\phi_{\mathcal{N}_i}$ and conditioned on $w_i^{\text{in}}$. As aforementioned, we approximately have $q_i^{\phi_i}(o_i, w_i^{\text{out}}) = v_i^{\pi_i}(o_i; w_i^{\text{in}})$. We can see that the action-value function of $\phi_i$ is equivalent to the value function of $\pi_i$. That said, we can use a single network to approximate these two functions simultaneously. For a deterministic low-level policy, the high-level and low-level policies can share a same action-value function. In the current instantiation of LToS, we use DDPG (Lillicrap et al., 2016) for the high-level policy and DGN (Jiang et al., 2020) (Q-learning) for the low-level policy. Thus, the Q-network of DGN also serves the critic of DDPG, and the gradient of $w_i^{\text{in}}$ is calculated based on the maximum Q-value of $a_i$. More discussions about training LToS and the detailed training algorithm are available in Appendix A.1.

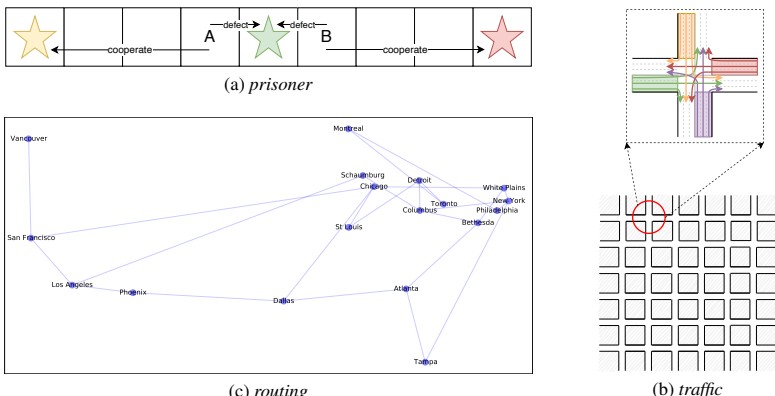

(a) *prisoner*

(c) *routing*

(b) *traffic*

Figure 2: Three experimental scenarios: (a) *prisoner*, (b) *traffic*, and (c) *routing*.

## 5 EXPERIMENTS

For the experiments, we adopt three scenarios depicted in Figure 2. *Prisoner* is a grid game about social dilemma that easily measures agents' cooperation, while *traffic* and *routing* are real-world scenarios of networked MARL. We obey the principle of networked MARL that only allows communication in neighborhood as Jiang et al. (2020); Chu et al. (2020).

To illustrate the reward sharing scheme each agent learned, we use a simple indicator: *selfishness*, the reward proportion that an agent chooses to keep for itself. For ablation, we keep the sharing weights fixed for each agent, named fixed LToS. Throughout the experiments, we additionally compare with the baselines including DQN, DGN, QMIX and two methods for networked MARL, *i.e.*, ConseNet (Zhang et al., 2018) and NeurComm (Chu et al., 2020), both of which take advantage of recurrent neural network (RNN) for the partially observable environment (Hausknecht & Stone, 2015). To maximize the average global reward directly, we specially tune the reward shaping factor of other baselines in *prisoner* and introduce QMIX as a centralized baseline in *traffic* and *routing*. Moreover, as DGN is the low-level policy of LToS, DGN also serves the ablation of LToS without reward sharing.

### 5.1 PRISONER

We use *prisoner*, a grid game version of the well-known matrix game *prisoner's dilemma* from Sodomka et al. (2013) to demonstrate that LToS is able to learn cooperative policies to achieve the global optimum (*i.e.*, maximize globally averaged return). As illustrated in Figure 2a, there are two agents A and B that respectively start on two sides of the middle of a grid corridor with *full* observation. At each timestep, each agent chooses an action *left* or *right* and moves to the corresponding adjacent grid, and each timestep every action incurs a cost $-0.01$. There are three goals, two goals at both ends and one in the middle. The agent gets a reward $+1$ for reaching the goal. The game ends once some agent reaches a goal or two agents reach different goals simultaneously. This game resembles prisoner's dilemma: going for the middle goal ("defect") will bring more rewards than the farther one on its side ("cooperate"), but if two agents both adopt that, a collision occurs and only one of the agents wins the goal with equal probability. On the contrary, both agents obtain a higher return if they both "cooperate", though it takes more steps.

Figure 3 illustrates the learning curves of all the models in terms of average return. Note that for all three scenarios, we represent the average of three training runs with different random seeds by solid lines and the min/max value by shadowed areas. As a result of self-interest optimization, DQN converges to the "defect/defect" Nash equilibrium where each agent receives an expected reward about $0.5$. So does DGN since it only aims to take advantage of its neighbors' observations while *prisoner* is a fully observable environment already. ConseNet agents sometimes choose to cooperate by building a consensus on average return at the beginning, but it is unstable

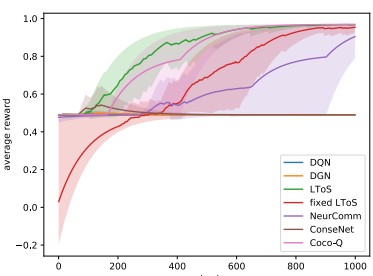

Figure 3: Learning curves in *prisoner*.

and abandoned subsequently. Given a hand-tuned reward shaping factor to direct agents to maximize average return, NeurComm and fixed LToS agents are able to cooperate eventually. However, they converge much slower. Coco-Q (Sodomka et al., 2013) and LToS outperform all other methods. As a modified tabular Q-learning method, Coco-Q introduces the coco value (Kalai & Kalai, 2010) as a substitute for the expected return in the Bellman equation and regards the difference as transferred reward. However, it is specifically designed for some games, and it is hard to be extended beyond two-player games. LToS can learn the reward sharing scheme where one agent at first gives all the reward to the other so that both of them are prevented from "defect", and thus achieve the best average return quickly. By *prisoner*, we verify LToS can escape from local optimum by learning to share reward.

## 5.2 TRAFFIC

In *traffic*, as illustrated in Figure 2b , we aim to investigate the capability of LToS in dealing with highly dynamic environment through reward sharing. We adopt the same problem setting as in (Wei et al., 2019). In a road network, each agent serves as traffic signal control at an intersection. The observation of an agent consists of a one-hot representation of its current phase (directions for red/green lights) and the number of vehicles on each incoming lane of the intersection. At each timestep, an agent chooses a phase from the pre-defined phase set for the next time interval, *i.e.*, 10 seconds. The reward is set to be the negative of the sum of the queue lengths of

Table 1: Statistics of traffic flows

| Time (second) | Arrival Rate (vehicles/s) |
|---|---|
| $0 - 600$ | 1 |
| $600 - 1,200$ | 1/4 |
| $1,200 - 1,800$ | 1/3 |
| $1,800 - 2,400$ | 2 |
| $2,400 - 3,000$ | 1/5 |
| $3,000 - 3,600$ | 1/2 |

all approaching lanes at current timestep. The global objective is to minimize average travel time of all vehicles in the road network, which is equivalent to minimizing the sum of queue lengths of all intersections over an episode (Zheng et al., 2019). The experiment was conducted on a traffic simulator, CityFlow (Zhang et al., 2019). We use a $6 \times 6$ grid network with 36 intersections. The traffic flows were generated to simulate dynamic traffic flows including both peak and off-peak period, and the statistics is summarized in Table 1.

Table 2: Average travel time of all the models in *traffic*

| DQN | DGN | fixed LToS | LToS | NeurComm | ConseNet | QMIX |
|---|---|---|---|---|---|---|
| 118.75 | 110.59 | 113.83 | **98.57** | 106.53 | 111.18 | 596.52 |

Figure 4 shows the learning curves of all the models in terms of average travel tiem of all vehicles in logarithmic form. The performance after convergence is summarized in Table 2, where LToS outperforms all other methods. LToS outperforms DGN, which demonstrates the reward sharing scheme learned by the high-level policy indeed helps to improve the cooperation of agents. Without the high-level policy, *i.e.*, given fixed sharing weights, fixed LToS does not perform well in dynamic environment. This indicates the necessity of the high-level policy. Although NeurComm and ConseNet both take advantage of RNN for partially observable environments, LToS still outperforms these methods, which verifies the great improvement

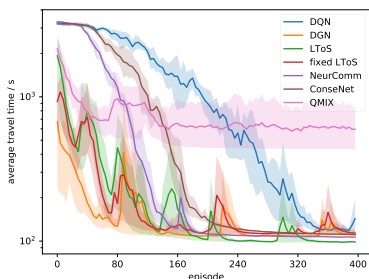

Figure 4: Learning curves in *traffic*.

of LToS in networked MARL. QMIX shows apparent instability and is confined to suboptimality (Mahajan et al., 2019). Specifically, in the best episode, QMIX tries to release traffic flows from one direction while stopping flows from the other all the time.

We visualize the variation of *selfishness* of all agents during an episodes in Figure 5 and 6. Figure 5 depicts the temporal variance of selfishness for each agent. For most agents, there are two valleys occurred exactly during two peak periods (*i.e.*, $0 - 600$s and $1,800 - 2,400$s). This is because for heavy traffic agents need to cooperate more closely, which can be induced by being less selfish. We can see this from the fact that selfishness is even lower in the second valley where the traffic is even heavier (*i.e.*, 2 *vs.* 1 vehicles/s). Therefore, this demonstrates that the agents learn to adjust their extent of cooperation to deal with dynamic environment by controlling the sharing weights. Figure 6 shows the spatial pattern of selfishness at different timesteps, where the distribution of agents is the same as the road network in Figure 2b. The edge and inner agents tend to have very

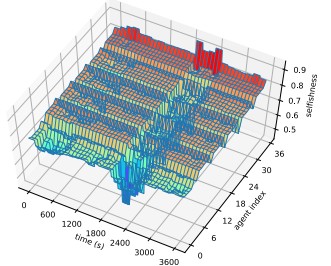

Figure 5: Temporal pattern of *selfishness*

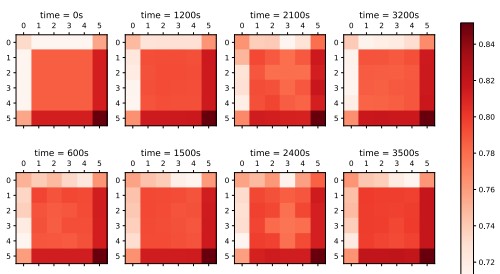

Figure 6: Spatial pattern of *selfishness*

different selfishness. In addition, inner agents keep their selfishness more uniform during off-peak periods, while they diverge and present cross-like patterns during peak periods. This shows that handling heavier traffic requires more diverse reward sharing schemes among agents to promote more sophisticated cooperation.

### 5.3 ROUTING

Packet routing is regarded as a complex problem in distributed computer networks. Here is a simplified version of the problem. A network consists of multiple routers with a stationary network topology. Data packets come into the network (started at a router) following the Poisson distribution, and the arrival rate varies during an episode as summarized in Table 3. Each router has a FIFO queue as the packet buffer. For simplicity, we assume that each queue has

Table 3: Statistics of packet flow

| Time (timestep) | Arrival Rate (packets/timestep) |
|---|---|
| 0 − 100 | 1 |
| 100 − 200 | 10/3 |
| 200 − 300 | 1 |

unlimited volume allowance, and each packet has a size same as each link's bandwidth. At every timestep, each router observes the data packets in the queue and incoming links as well as indices of neighboring routers, forwards the first packet in the FIFO to the selected next hop, and obtains a reward which is the negative of the queue length. The transmission time of a packet over a link is proportional to the geographic distance, and the packet will be stored after arriving at the next hop unless it reaches the destination. The delay of a packet is the sum of timesteps spent at the routers and on the links. The goal of packet routing is to send the packets to their destinations through hop-by-hop transmissions with minimum average delay. Compared with *traffic*, *routing* is a more fine-grained task, because it requires specific control for each data packet.

In the experiment, we choose a real network topology: IBM backbone network of 18 vertices that each works for a city in North America (Knight et al., 2011) and the topology is depicted in Figure 2c, where each edge consists of two unidirectional links and varies considerably in distance. We assume that each router helps with loopback detection while forwarding. Figure 7 illustrates the learning curves of all the models in terms of average delay, and their performance after convergence in terms of throughout and delay is also summarized in Table 4. NeurComm, ConseNet and QMIX are not up to this task and may need much more episodes to converge. By learning proper reward sharing, LToS outperforms all other base-

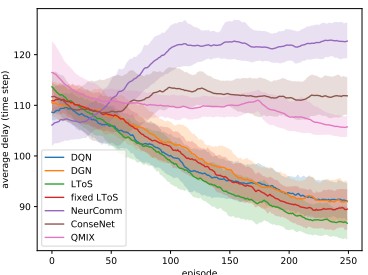

Figure 7: Learning curves in *routing*.

lines in terms of both metrics. Compared to *traffic*, *routing* additionally considers the heterogeneous network topology. Therefore, the experimental results also verify the capability of LToS of handling both temporal and spatial heterogeneity in networked MARL.

Table 4: Performance of all models in *routing*: throughput (packets) and average delay (timesteps)

| | DQN | DGN | fixed LToS | LToS | NeurComm | ConseNet | QMIX |
|---|---|---|---|---|---|---|---|
| throughput | 297.69 | 299.24 | 304.99 | **311.50** | 116.53 | 185.32 | 218.79 |
| delay | 91.06 | 90.96 | 89.50 | **86.71** | 122.68 | 111.87 | 105.74 |

## 6 CONCLUSION

In this paper, we proposed LToS, a hierarchically decentralized method for networked MARL. LToS enables agents to share reward with neighbors so as to encourage agents to cooperate on the global objective. For each agent, the high-level policy learns how to share reward with neighbors to decompose the global objective, while the low-level policy learns to optimize local objective induced by the high-level policies in the neighborhood. Experimentally, we demonstrate that LToS outperforms existing methods in both social dilemma and two networked MARL scenarios.

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

# A APPENDIX

## A.1 ALGORITHM AND DISCUSSIONS

As a hierarchically decentralized MARL method, LToS brings some challenges for training. Algorithm 1 presents the training algorithm of LToS.

**Selfishness Initializer.** On the basis of a straightforward idea that one should generally focus more on its own reward than that of others when optimizing its own policy, the initial output of each high-level policy network is supposed to be higher on the sharing weight of its own than others. We

---

**Algorithm 1** LToS

---

1: Initialize $\phi_i$ parameterized by $\theta_i$ and $\pi_i$ parameterized by $\mu_i$ for each agent $i$ ($\phi_i$ is learned using DDPG and $\pi_i$ is learned using DGN, where they share the Q-network)
2: **for** episode = 1 to max-training-round **do**
3:     Initialize a random process $\mathcal{X}^w$ for $w$-action exploration
4:     Initialize a random process $\mathcal{X}^a$ for $a$-action exploration
5:     **for** max-episode-length **do**
6:         **for** each agent $i$ **do**
7:             $w_i^{\text{out}} \leftarrow \phi_i(o_i) + \mathcal{X}^w$
8:             $a_i \leftarrow \pi_i(o_i; w_i^{\text{in}}) + \mathcal{X}^a$
9:             Execute action $a_i$, obtain original reward $r_i$, and transition to $o_i'$
10:            Exchange $w_i^{\text{out}}$ and $r_i$, and get $w_i^{\text{in}}$ and $r_i^w$
11:            Store $(o_i, w_i^{\text{in}}, a_i, r_i^w, o_i')$
12:         **end for**
13:         **if** time to update **then**
14:             **for** each agent $i$ **do**
15:                 Sample a minibatch $D = \{(o_i, w_i^{\text{in}}, a_i, r_i^w, o_i')\}$ from replay buffer $\mathcal{B}_i$
16:                 Exchange $w_i^{\text{out}'} \leftarrow \phi_i'(o_i')$ and get $w_i^{\text{in}'}$
17:                 Set $y_i \leftarrow r_i^w + \gamma q_i^{\pi_i'}(o_i', a_i'; w_i^{\text{in}'})|_{a_i' = \pi_i'(o_i'; w_i^{\text{in}'})}$
18:                 Update $\pi_i$ by $\nabla_{\mu_i} \frac{1}{|D|} \sum_{(o_i, w_i^{\text{in}}, a_i, r_i^w, o_i') \in D} (y_i - q_i^{\pi}(o_i, a_i; w_i^{\text{in}}))^2$
19:                 Exchange $w_i^{\text{out}} \leftarrow \phi_i(o_i)$ and get $w_i^{\text{in}}$
20:                 Calculate the gradient $g_i^{\text{in}} = \nabla_{w_i^{\text{in}}} q_i^{\pi_i}(o_i, \arg\max_{a_i} q_i^{\pi_i}; w_i^{\text{in}})$
21:                 Exchange $g_i^{\text{in}}$ and get gradient $g_i^{\text{out}}$ for $w_i^{\text{out}}$
22:                 Update $\theta_i$ by $\frac{1}{|D|} \sum_{o_i \in D} (\nabla_{\theta_i} \phi_i(o_i))^{\mathsf{T}} g_i^{\text{out}}$
23:                 Softly update $\theta_i'$ and $\mu_i'$: $\theta_i' \leftarrow \tau\theta_i + (1-\tau)\theta_i'$ and $\mu_i' \leftarrow \tau\mu_i + (1-\tau)\mu_i'$
24:             **end for**
25:         **end if**
26:     **end for**
27: **end for**

---

choose to predetermine the initial selfishness to learn the high-level policy effectively. However, with normal initializers, the output of the high-level policy network will be evenly distributed initially. Therefore, we use a special *selfishness initializer* for each high-level policy network instead. As we use the softmax to produce the weights, which guarantees the constraint: $\sum_{j \in \mathcal{N}_i} w_{ij} = 1, \forall i \in \mathcal{V}$, we specially set the bias of the last fully-connected layer so that each decentralized high-level policy network tends to keep for itself the same reward proportion as the given selfishness initially. The rest of reward is still evenly distributed among neighbors. LToS learns started from such initial weights, while fixed LToS uses such weights throughout each experiment. Moreover, we use grid search to find the best selfishness for fixed LToS in *traffic* and *routing*. For *prisoner* we deliberately set the selfishness to $0.5$ so that fixed LToS directly optimizes the average return.

**Unified Pseudo Random Number Generator.** LToS is learned in a decentralized manner. This incurs some difficulty for experience replay. As each agent $i$ needs $w_i^{\text{in}}$ to update network weights for both high-level and low-level policies, it should sample from its buffer a batch of experiences where each sampled experience should be synchronized across the batches of all agents (*i.e.*, the experiences should be collected at a same timestep). To handle this, all agents can simply use a unified pseudo random number generator and the same random seed.

**Different Time Scales.** As many hierarchical RL methods do, we set the high-level policy to running at a slower time scale than the low-level one. Proposition 4.1 still holds if we expand $v_i^{\boldsymbol{\pi}}$ for more than one step forward. Assuming the high-level policy runs every $M$ timesteps, we can fix $w_i^{\text{out},t} = w^{\text{out},t+1} = \cdots = w^{\text{out},t+M-1}$. $M$ is referred to as *action interval* in Table 6.

**Infrequent Parameter Update with Small Learning Rate.** Based on the continuity of $\boldsymbol{w}$, a small modification of $\phi$ means a slight modification of local reward functions, and will intuitively result

Table 5: Hyperparameters for DQN and DGN (also serves as the low-level policy network of LToS)

| Hyperparamater | *Prisoner* | *Traffic* | *Routing* |
|---|---|---|---|
| sample size | 10 | 1,000 | 10 |
| batch size | 10 | 20 | 10 |
| buffer capacity | 200,000 | 10,000 | 200,000 |
| $\epsilon$/decay/minimum value | 0.8/1/0.8 | 0.4/0.9/0.05 | 0.2/0.98/0 |
| initializer | random normal | random normal | random normal |
| optimizer | Adam | Adam | Adam |
| learning rate | 1e-3 | 1e-3 | 1e-3 |
| $\gamma$ | 0.99 | 0.8 | 0.99 |
| $\tau$ for soft update | 0.1 | 0.1 | 0.1 |
| # MLP units | 32 | 32 | 128 |
| MLP activation | ReLU | ReLU | ReLU |
| # encoder MLP layers | 2 | 2 | 2 |
| # attention heads for DGN | 4 | 1 | 8 |
| # convolutional layers for DGN | 1 | 1 | 1 |

Table 6: Hyperparameters for the high-level policy network of LToS

| Hyperparamater | *Prisoner* | *Traffic* | *Routing* |
|---|---|---|---|
| update frequency | 1 step | 5 episodes | 20 episodes |
| action interval | 1 step | 15 steps | 30 steps |
| sample size | 2,000 | 1,000 | 2,000 |
| batch size | 32 | 20 | 32 |
| noise for exploration | $\epsilon$ + Gaussian | OU | OU |
| noise parameter | $\epsilon = 0.8, \sigma = 1$ | $\sigma = 0.25\epsilon$ | $\sigma = 0.025\epsilon$ |
| initializer | selfishness | selfishness | selfishness |
| initial selfishness | 0.5 | 0.8 | 0.9 |
| optimizer | SGD | SGD | SGD |
| learning rate | 1e-1 | 1e-3 | 1e-3 |
| last MLP layer activation | softmax | softmax | softmax |

Table 7: Hyperparameters for NeurComm, ConseNet and QMIX

| Hyperparamater | *Prisoner* | *Traffic* | *Routing* |
|---|---|---|---|
| initializer | orthogonal | orthogonal | orthogonal |
| optimizer | RMSProp | RMSProp | RMSProp |
| learning rate | 5e-3 | 5e-4 | 5e-4 |
| # MLP units | 20 | 16 | 128 |
| MLP activation | ReLU | ReLU | ReLU |
| # cell state units | 20 | 16 | 128 |
| # hidden state units | 20 | 16 | 128 |
| RNN type for NeurComm and ConseNet | LSTM | LSTM | LSTM |
| RNN type for QMIX | - | GRU | GRU |
| hypernetwork layer1 units for QMIX | - | $36 \times 16$ | $18 \times 128$ |
| hypernetwork layer2 units for QMIX | - | 16 | 128 |
| $\alpha$ for NeurComm | 1 | 0.1 | 0.05 |

in an equally slight modification of the low-level value functions. This guarantees the low-level policies are highly reusable.

## A.2 HYPERPARAMATERS

Table 5 summarizes the hyperparameters of DQN and DGN that also serves as the low-level network of LToS. We follow many of the original DGN in *prisoner* and *routing*, but choose the setting of Wei et al. (2019) in *traffic* for consistency. Table 6 summarizes the hyperparameters of the high-level network of LToS, which are different from the low-level network. Table 7 summarizes the

hyperparameters of NeurComm and ConseNet, which adhere to the implementation (Chu et al., 2020). In addition, for tabular Coco-Q, the step-size parameter is 0.5. We adopt soft update for both high-level and low-level networks and use an Ornstein-Uhlenbeck Process (abbreviated as OU) for high-level exploration.

Both fixed LToS and NeurComm exploit static reward shaping, but they adopt different reward shaping schemes which are hard to compare directly. We consider a simple indicator: Self Neighbor Ratio (SNR), the ratio of reward proportion that an agent chooses to keep for itself to that it obtains from a single neighbor. As the rest reward is evenly shared with neighbors in LToS, for each agent $i$, we have $\text{SNR} = {}^{\text{selfishness}}/_{\text{1-selfishness}} \times (|\mathcal{N}_i| - 1)$ for LToS, and $\text{SNR} = {}^{1}/_{\alpha}$ for NeurComm where $\alpha$ is the spatial discount factor. We adjust the initial selfishness and $\alpha$ to set the SNR of both methods at the same level for fair comparison.

