# OpenReview forum: "Learning to Share in Multi-Agent Reinforcement Learning"
_ICLR.cc/2021/Conference — Reject_

### Official Review · AnonReviewer1 · 2020-10-21
**Some interesting ideas, but issues with formalizing the problem setting, theory and unconvincing results.**

**Rating:** 4
**Confidence:** 4

**Review:**

Update: I appreciate the detailed replies to my questions. Indeed, some of the points I raised were addressed well and the paper updated accordingly.

However, some new concerns were also raised by the replies:
- Using 3 seeds for the experimental evaluation is an extremely questionable evaluation protocol. There is no way to know if any of the results are going to hold up.
- It's also clear now that none of the experiments are comparing to benchmark numbers from other publications. It would have been more confidence inspiring if the method was tested on a set of tasks where external benchmarks have already been established.
- This is particularly true for the new results that were added to the paper, e.g. the QMIX results. It's difficult to make sense of them and the instability points towards a potential hyperparameter issue.
- All baselines for the 'prisoners' case should at least compare to the fully cooperative case of adding up the rewards. Comparing to a DQN baseline that maximizes individual rewards is a red herring.
- It's odd that all experiments require less than 1000 episodes to train. This is very unusual for challenging multi-agent RL problems. It would be great to understand if the main selling point of LToS is sample-complexity/learning speed or if there is something else going on.

I also agree with the concern raised by other reviewers that the paper is currently not positioned clearly.
All things considered, I believe my score is still appropriate for the paper. However, I also believe that a future version of the paper with clarified positioning and more thorough experimental evaluation could make for a compelling contribution.

Original review:
==========
-"Obviously, CTDE cannot address such problems due to the curse of dimensionality.". CTDE means that there is the *option* to use centralized information at training time. Clearly, some ways of using centralized information will scale better than others and claiming that none of them scale is simply unfounded.

-"One is that the reward function.. tragedy of the commons.". I am struggling to make sense of this paragraph. Please work on the clarity of the writing.

-"However, they are learned in a centralized way and hence not scalable." These methods have been scaled to large numbers of agents in complex environments. Please provide citations when making a claim that something doesn't scale. For example, the "The StarCraft Multi-Agent Challenge", Samvelyan et al 2020, includes results for numbers of agents comparable to the largest experiments in this paper.

-"Moreover, the observation of each agent oi ∈ Oi can be enhanced to the Cartesian product of agent i and its neighbors (Jiang et al.,
2020) or the observation history (Chu et al., 2020)". I don't follow this. If the observation of each agent includes the observation of all neighbors (which includes the observation of their neighbors), then shouldn't everyone observe everything?

-Equation (1) is wrong. The left-hand side conditions on 'oi_', but the right-hand side conditions on 's'. This also affects all following equations.

-"The simple way to optimize the global objective is that each agent maximizes its own expected return, which is known as Markov game. ". This is wrong. When each agent optimizes their own expected return this is typically not a means of optimizing the global objective.

-". In networked MARL, as the reward of an agent is assumed to depend on the actions of neighbors, we allow reward
sharing between neighboring agents": The reward function also depends on the global state, 's', which is a function of the joint action of all of the agents. So this local reward sharing seems clearly insufficient in general.

- Eqn 6 to 15: This proof seems unnecessarily cumbersome. W only redistributes the rewards, so the sum of total rewards is unchanged, qed.

-"Unlike existing hierarchical RL methods, we can directly construct the value function and action value function of φ based on the value function of π at each agent.": Constructing the value function isn't really the problem, but approximating and learning it is challenging.

Theory:
-4.3: "Each vertex has its own local policy φij (wij |oi), and we can verify their independence by means of Markov Random Field." This is not clear to me. Furthermore, given that the transition function conditions on the joint action and that the reward function depends on the central state, this seems wrong. Unless I am mistaken, the dependency on the central state should break any locality assumptions.

Experiments:
- The results on the prisoner's dilemma are misleading. Clearly, if there is an ability to change the reward functions of individual agents (which is assumed by LToS), there is no more social dilemma. As such, only baselines that maximize the total reward are credible comparisons (and seem to be missing completely).
- The "traffic" and "ROUTING" experiments seem more interesting. A few caveats: None of the results include uncertainty estimates. It is furthermore unclear, how many seeds were used. Furthermore, the fixed LToS baseline ("For ablation, we keep the sharing weights fixed for each agent, named fixed LToS") seems odds. Did you try a baseline where all agents simply share their reward equally with their neighbors? Also, centralized baselines are missing. E.g: https://arxiv.org/pdf/1910.00091.pdf.
- In "ROUTING" Fixed LToS (ie. not learning to share) and LToS seem indistinguishable.

---

> ### Author Response · Authors · 2020-11-22
> **Responses to Reviewer 1: part 2**
>
> > "Unlike existing hierarchical RL methods, we can directly construct the value function and action value function of $\boldsymbol{\phi}$ based on the value function of $\boldsymbol{\phi}$ at each agent.": Constructing the value function isn't really the problem, but approximating and learning it is challenging."
>
> Yes, it is no problem to construct a new value function. What we want to avoid is learning the new value function. By this property, we are able to reuse the value function of the low-level policy as we do in the current implementation of LToS.
>
> > The results on the prisoner's dilemma are misleading. Clearly, if there is an ability to change the reward functions of individual agents (which is assumed by LToS), there is no more social dilemma. As such, only baselines that maximize the total reward are credible comparisons (and seem to be missing completely).
>
> Yes, for credible comparisons, we exactly use fixed LToS and NeurComm as baselines that maximize the total return. As can be seen in Appendix, we specially set their selfishness and $\alpha$ to direct agents to maximize average return. As a result, they are able to cooperate eventually but converge much slower.
>
> > None of the results include uncertainty estimates. It is furthermore unclear how many seeds were used. Furthermore, the fixed LToS baseline ("For ablation, we keep the sharing weights fixed for each agent, named fixed LToS") seems odds. Did you try a baseline where all agents simply share their reward equally with their neighbors? Also, centralized baselines are missing. E.g: https://arxiv.org/pdf/1910.00091.pdf
>
> First, we do not include uncertainty estimates since the test does not contain any random factor. That is because we generated the vehicle and data packet flows once and kept that throughout the whole experiment, partly because vehicle routes have to be fixed in CityFlow. We use three random seeds and show in the tables the average results over them. For fixed LToS, we fix the selfishness and all neighbors have the same sharing weight. For example, if we fix the selfishness of each agent at 0.2 and each agent has four neighbors, then each neighbor gets 0.2. As we performed the grid search to find the best selfishness for fixed LToS in traffic and routing, the experiment should cover the case "all agents simply share their reward equally with their neighbors." For centralized baselines, we additionally compared with QMIX, we can see that in the revision QMIX does not perform well in *traffic*.
>
> > in "ROUTING" Fixed LToS (ie. not learning to share) and LToS seem indistinguishable.
>
> We believe that the figure could show some difference. We admit that for *routing*, the gap between LToS and other baselines is smaller compared to *prisoner* and *traffic*.

---

> ### Author Response · Authors · 2020-11-22
> **Responses to Reviewer 1: part 1**
>
> > "Obviously, CTDE cannot address such problems due to the curse of dimensionality." CTDE means that there is the option to use centralized information at training time. Clearly, some ways of using centralized information will scale better than others and claiming that none of them scale is simply unfounded.
>
> > "However, they are learned in a centralized way and hence not scalable." These methods have been scaled to large numbers of agents in complex environments. Please provide citations when making a claim that something doesn't scale. For example, the "The StarCraft Multi-Agent Challenge", Samvelyan et al 2020, includes results for numbers of agents comparable to the largest experiments in this paper.
>
> We have revised these claims to make them accurate. We position LToS at a particular line of research of networked MARL, where CTDE methods may not easily scale up with the number of agents as empirically demonstrated in Qu et al., 2020a, including MADDPG, QMIX. We also additionally performed the experiment of QMIX in 1) *traffic*, and it shows QMIX does not perform well as illustrated in Figure 4 and Table 2, and 2) *routing*, and it shows QMIX does not perform well either as illustrated in Figure 7 and Table 4.
>
> > "One is that the reward function... tragedy of the commons." I am struggling to make sense of this paragraph. Please work on the clarity of the writing.
>
> > "The simple way to optimize the global objective is that each agent maximizes its own expected return, which is known as Markov game. " This is wrong. When each agent optimizes their own expected return this is typically not a means of optimizing the global objective.
>
> We have revised Introduction and Background to make them precise and clear.
>
> > "Moreover, the observation of each agent $o_i \in O_i$ can be enhanced to the Cartesian product of agent i and its neighbors (Jiang et al., 2020) or the observation history (Chu et al., 2020)". I don't follow this. If the observation of each agent includes the observation of all neighbors (which includes the observation of their neighbors), then shouldn't everyone observe everything?"
>
> The observations of neighbors are obtained by communication, not naturally, and $o_i$ denotes the information available after communication. We have revised this part to make it clear.
>
> > "In networked MARL, as the reward of an agent is assumed to depend on the actions of neighbors, we allow reward sharing between neighboring agents": The reward function also depends on the global state, 's', which is a function of the joint action of all of the agents. So this local reward sharing seems clearly insufficient in general.
>
> Although the joint action of all agents determines the state transition, hence state distribution, only the actions in the neighborhood of an agent determine its reward at a particular state. Therefore, it is easy for an agent to learn to improve the return by sharing reward with neighbors, since the change of actions of neighbors directly affects the reward. However, the agents outside the neighborhood can only affect the return of the agent by the change of state distribution. It is hard for an agent to learn to amount the effect of reward sharing on the state distribution. Therefore, we consider only local reward sharing.
>
> > Equation (1) is wrong. The left-hand side conditions on 'o_i', but the right-hand side conditions on 's'. This also affects all following equations.
>
> > Theory: -4.3: "Each vertex has its own local policy φij (wij |oi), and we can verify their independence by means of Markov Random Field." This is not clear to me. Furthermore, given that the transition function conditions on the joint action and that the reward function depends on the central state, this seems wrong. Unless I am mistaken, the dependency on the central state should break any locality assumptions.
>
> We have revised the paper to clearly present the mathematical workflow behind LToS. Specifically, in (1), $o_i$ should be $s$, and in Proposition 2, $\phi_{ij} (w_{ij} |o_i)$ should be $\phi_{ij} (w_{ij} |s)$. The main difference is we deferred the approximation of state by observations (or history) to the end, which makes the mathematical part simple and clear.
>
> > Eqn 6 to 15: This proof seems unnecessarily cumbersome. W only redistributes the rewards, so the sum of total rewards is unchanged, qed.
>
> Yes. We just want to put it more rigorous.

---

### Official Review · AnonReviewer4 · 2020-10-21
**Needs some improvement**

**Rating:** 4
**Confidence:** 4

**Review:**

The paper addresses multi-agent RL problems by presenting a decentralized approach where the agents learn to share their reward with their neighbors. In this method, a high-level policy determines a weight vector for weighting the reward of neighboring agents, and then each agent learns their own independent policy. The learning is thus conducted locally in a partially connected network toward a common goal and without the knowledge of global state and actions.

Overall, the approach is intuitive and interesting for decentralized learning in MARL tasks. However, I have some comments/questions for improving the paper that are summarized below.  Hence, I vote to reject at this stage.

Pros:
+ Intuitive design of communication among agents in decentralized setting
+ Clever adaption of algorithms
+ Well written paper and properly organized

Comments:
- The contribution of the paper is mainly in formulating the problem in the actor-critic setup of DDPG method which leads to a limited novelty.

- A key concern about the paper is how to decompose the reward in the first place. The paper aims at optimizing a global objective and assumes (also in the propositions) that this objective has additive connection with the decentralized rewards. Nevertheless, this is a strong assumption, particularly in real-world applications. A global reward can be decomposed into summation of smaller rewards, but not necessarily the other way around. As long as there is a global objective, we need a way to distribute the reward among the agents via learning or reward reshaping (or even manually). How can we properly define the reward of each agent in such scenarios?

- It is also unclear what is the benefit of sharing only with the neighbors. The method learns a weight vector of size |N_i| for every agent. Does it make a difference in the architecture/algorithm if we learn the weights of all the other agents (size |N|) instead?

- Formulating the weights as finite discrete values looks unnatural. If the method is designed for continuous action space, it is expected to have the notations to be continuous as well. Can we just simply convert the summations into integration in the propositions!?

- The authors claim that the problem with the related work is that they can not scale up with the number of agents. However, there is no (empirical) support that how the proposed approach deals with large-scale problems.

- In general, the experiments are small and based on simulation, and simulated scenarios are not considered real-world (which is claimed otherwise in the paper). I would recommend to incorporate more supportive empirical evaluation.


Minor:
What is \phi_{-i} in eq 17

---

> ### Author Response · Authors · 2020-11-22
> **Responses to Reviewer 4**
>
> > The contribution of the paper is mainly in formulating the problem in the actor-critic setup of DDPG method which leads to a limited novelty.
>
> Our main contribution is how to learn reward sharing so as to maximize the average return of all agents in networked MARL. LToS is a new hierarchical MARL framework rather than an actor-critic setup, so it is not restricted to current instantiation of DDPG+DGN but can be realized by diverse combinations of methods.
>
> > A key concern about the paper is how to decompose the reward in the first place. The paper aims at optimizing a global objective and assumes (also in the propositions) that this objective has additive connection with the decentralized rewards. Nevertheless, this is a strong assumption, particularly in real-world applications. A global reward can be decomposed into summation of smaller rewards, but not necessarily the other way around. As long as there is a global objective, we need a way to distribute the reward among the agents via learning or reward reshaping (or even manually). How can we properly define the reward of each agent in such scenarios?
>
> In networked MARL, we do not decompose the reward, but each agent naturally has an individual reward, following the setting of Zhang et al. (ICML’18), Qu et al. (NeurIPS’19), Chu et al. (ICLR’20), Qu et al. (NeurIPS’20). For example, in traffic, the reward of each agent is the negative of the queue lengths. In networked MARL, the main problem is just how to advocate cooperating on the objective of maximizing the average of cumulative rewards of all agents, which serves as the global objective.
>
> > It is also unclear what is the benefit of sharing only with the neighbors. The method learns a weight vector of size |N_i| for every agent. Does it make a difference in the architecture/algorithm if we learn the weights of all the other agents (size |N|) instead?
>
> In networked MARL, a common assumption is that the reward of each agent just depends on its action and the actions of its neighbors (Qu et al., 2020a). Thus, LToS only learns to share reward with neighbors, which is also limited by communication. For the case of |N|, each agent needs to output the weight for all other agents and take as input the weights from all other agents, and thus it becomes more centralized, which contradicts to the decentralized learning of networked MARL.
>
> > Formulating the weights as finite discrete values looks unnatural. If the method is designed for continuous action space, it is expected to have the notations to be continuous as well. Can we just simply convert the summations into integration in the propositions!?
>
> Yes, we use summations only to ease the presentation.
>
> > The authors claim that the problem with the related work is that they cannot scale up with the number of agents. However, there is no (empirical) support that how the proposed approach deals with large-scale problems.
>
> Actually, we do not argue that our method is very scalable while these factorization methods are totally not. We have amended the claims. Networked MARL focuses on decentralized learning as well as maximizing average return of all agents, while factorization methods are centralized training and focus the case all agents share a reward. Factorization methods can certainly be applied networked MARL. But in the literature it is empirically shown that QMIX performs poorly in large-scale networked MARL (Qu et al., 2020a). We also additionally performed the experiment of QMIX in 1) *traffic*, and it shows QMIX does not perform well as illustrated in Figure 4 and Table 2, and 2) *routing*, and it shows QMIX does not perform well either as illustrated in Figure 7 and Table 4.
>
> > In general, the experiments are small and based on simulation, and simulated scenarios are not considered real-world (which is claimed otherwise in the paper). I would recommend to incorporate more supportive empirical evaluation.
>
> We have removed these in the revision.
>
> > Minor: What is $\phi_{-i}$ in eq 17
>
> As stated in Equation (2), it means the joint policy of all agents except agent $i$.

---

### Official Review · AnonReviewer2 · 2020-10-27
**An interesting paper on learning to share rewards**

**Rating:** 8
**Confidence:** 3

**Review:**

Summary
The paper considers the cooperative MARL setting where agents get local rewards and they are interconnected as a graph where neighbors can communicate. The paper specifically considers the communication of reward sharing, that is, an agent shares (part of) its reward to its neighbors, such that each agent optimizes its local reward plus rewards from its neighbors. This motivates a bi-level optimization framework where the high-level policy decides how the rewards are shared and the low-level policy locally optimizes the shared rewards given the high-level’s decision. The paper’s flow motivates such a framework well. The experimental results demonstrate the method’s effectiveness. I think it is a strong paper (accept), but my confidence is low due to the following confusions I have.

Comments/Questions

1. I have a high-level comment on the reward sharing mechanism. It seems that the proposed method does not support multi-hop sharing because rewards can only be shared to neighbors. Why is this single-hop sharing effective in the experiments? Is it because of domain-specific reasons, or it’s because that single-hop sharing is in principle equally effective, why?

2. The derivation of (18) using taylor expansion is unclear to me. Could the authors explain it with more details?

3. I don’t fully understand the proof of Proposition 4.2. Specifically, does “phi can be learned in a decentralized manner” mean that the *optimal* phi can be based on only the local observation for each agent, instead of based on global state? Could the authors comment on the approximation error induced by the mean-field approximation? Why the proof begins with phi_i based on o_i and ends with phi_i based on global state s.

4. In Equation (17) and (20), should phi^* be just phi (i.e. no * here)?

5. The low-level policy is to optimize the shared rewards. My understanding is that any (single-agent) RL algorithm can be used for optimizing the shared rewards, e.g. DQN, DDPG, A2C, etc. Why would the authors choose DGN, a rather less popular RL algorithm? Have the authors tried more popular algorithms as the low-level policy?

6. For fixed LToS,  how do we determine the fixed sharing weights?

---
Thanks for the response. I've increased my confidence.

---

> ### Author Response · Authors · 2020-11-22
> **Responses to Reviewer 2**
>
> > Why is this single-hop sharing effective in the experiments? Is it because of domain-specific reasons, or it is because that single-hop sharing is in principle equally effective, why?
>
> There is an implicit assumption in networked MARL that each agent can only perform single-hop communication in one timestep. That is, two-hop communication requires two timesteps, which could make communicated information outdated. But, how to deal with this is not the focus of this paper.
>
> > The derivation of (18) using taylor expansion is unclear to me. Could the authors explain it with more details?
>
> It is not suitable to be explained as a Taylor expansion. It is just alternative update as in DARTS (Liu et al. ICLR'19) to settle the problem of bi-level optimization. We have corrected this in the revision.
>
> > I don’t fully understand the proof of Proposition 4.2. Specifically, does “phi can be learned in a decentralized manner” mean that the optimal phi can be based on only the local observation for each agent, instead of based on global state? Could the authors comment on the approximation error induced by the mean-field approximation? Why the proof begins with $\phi_i$ based on $o_i$ and ends with $\phi_i$ based on global state s.
>
> Sorry for the confusion. We have revised the paper to address this. We have deferred the approximation of state by observations (or history) to the end, which makes the mathematical part simple and clear. About the approximation error, we are afraid that we could not give a generic quantitative error analysis at this stage, because it is extremely hard to model and analyze the error brought in by the reduction of action dependency.
>
> We notice that there are some works that study on the theoretical foundation and error analysis of MARL, but they usually rely on some strong and special assumptions, like exponential decay property and independent state (local observation)  transition (Qu et al. 2019, arXiv, 1912.02906) which do not hold in many real applications and thus become not general but limited (Qu et al. NeurIPS'20).
>
> > In Equation (17) and (20), should phi^* be just phi (i.e. no * here)?
>
> Typos, $\phi^*$ is supposed to be $\phi$ here. We have corrected this.
>
> > The low-level policy is to optimize the shared rewards. My understanding is that any (single-agent) RL algorithm can be used for optimizing the shared rewards, eg DQN, DDPG, A2C, etc. Why would the authors choose DGN, a rather less popular RL algorithm? Have the authors tried more popular algorithms as the low-level policy?"
>
> The understanding is correct. We consider our LToS more of a new hierarchical MARL framework than a new method, so it's not restricted to DDPG+DGN but can be realized by diverse combinations of methods. We currently employed DGN in our experimental because it is capable to handle communication while others (DQN, DDPG, A2C) are not, and it has shown its advantage over others like CommNet (Sukhbaatar et al. NeurIPS 2016).
>
> > For fixed LToS, how do we determine the fixed sharing weights?
>
> For *prisoner*, we set selfishness specially to average global return (i.e., 0.5). For *traffic* and *routing*, we used the best fixed selfishness found by grid search. We have made this clear in the revision.

---

### Official Review · AnonReviewer3 · 2020-10-28
**Learning to Share in Multi-Agent Reinforcement Learning**

**Rating:** 8
**Confidence:** 3

**Review:**

The paper present a new method, called LToS which enables agents to share rewards in MARL. Two levels of policies, high-level and low-level, determines rewards and optimize global objectives. Three diverse scenarios were used to test the performance of LToS compared to other baseline methods. LToS consistently outperforms other methods. In the second scenario, authors also show the need for high-level policy by introduction fixed LToS.

- At the end of Introduction, the sentence ‘LToS is easy to implement and currently realized by DDPG…’ can be misleading because of the word ‘realized’ and the fact that authors argue that LToS is a newly proposed method. Does this mean LToS simply combines DDPG and DGN?
- Do Figure 5 and 6 represent selfishness of agents when LToS is used?
- Minor editorial errors in Appendix

---

> ### Author Response · Authors · 2020-11-22
> **Responses to Reviewer 3**
>
> > At the end of Introduction, the sentence ‘LToS is easy to implement and currently realized by DDPG…’ can be misleading because of the word ‘realized’ and the fact that authors argue that LToS is a newly proposed method. Does this mean LToS simply combines DDPG and DGN? Do Figure 5 and 6 represent selfishness of agents when LToS is used?
>
> We consider our LToS more of a new hierarchical MARL framework than a new method, so it is not restricted to DDPG+DGN but can be realized by diverse combinations of methods. It is just "currently realized" by DDPG+DGN in the experiments. Yes, Figure 5 and 6 are trying to represent temporal and spatial pattern of agents' selfishness of LToS.

---

### Official Review · AnonReviewer5 · 2020-11-06
**On Positioning and Evaluation**

**Rating:** 3
**Confidence:** 4

**Review:**

The paper proposes a hierarchical multi-agent reinforcement learning method for the restricted communication setting and verifies the algorithm performance in a number of useful applications. The hierarchical approach to the networked MARL problem proves novel, effective, and interesting.

+ Strengths:

+ The work targets an arguably less explored area by focusing on the restrictions on inter-agent communication that may be present in realistic scenarios.

+ Evaluation setup is varied, explained in detail and visualized in an intuitive manner.

+ Niche is well-identified, and the contribution is clear.


- Major Concerns:

- The reviewer had issues positioning the paper among the different lines of research. Although the research gap itself is clear (scalable MARL methods in a restricted communication setting), it isn't obvious why and how relevant the cited works are. For example, mentioning VDN, QMIX, and QTRAN (which together are some of the latest works in the factorization methods) does not seem to serve any further purpose, as they are no longer compared quantitatively or qualitatively to LToS. The authors' claim that they are not scalable leads the reviewer to anticipate that LToS naturally is scalable, but there appears to be no evidence whatsoever presented in the latter sections of the paper to show, let alone prove, the superior scalability of LToS, with, for example, growing numbers of agents and training times.

-  Furthermore, some of the cited works have been left out at the evaluation stage, which leaves the reviewer puzzled as to which baselines LToS really hopes to outshine. The work needs some justification over why the following studies have not been compared to in the evaluation:

If the main strength of LToS lies in its capability to function effectively and efficiently in restricted communications setting, comparison to one or more of the following works should be of great advantage in illustrating that edge:
DIAL/RIAL by  Foerster 2016 - Learning to Communicate with Deep Multi-Agent Reinforcement Learning
BiCNet by Peng 2017 arXiv - Multiagent Bidirectionally-Coordinated Nets
CommNet by Sukhbaatar 2016 NeurIPS - Learning Multiagent Communication with Backpropagation
IC3Net by Singh 2019 ICLR - Learning When to Communicate at Scale in Multiagent Cooperative and Competitive Tasks
SchedNet by Kim 2019 ICLR - Learning to Schedule Communication in Multi-agent Reinforcement Learning

If the main strength of LToS lies in its capability to resolve selfishness and assign credits appropriately to bring about a harmonious cooperation in social dilemmas, analysis with respect to the this work should be helpful:
Eccles 2019 CoRR - Learning Reciprocity in Complex Sequential Social Dilemmas

It would be interesting to draw some parallels between LToS and BAD, as both draw inspiration from a hierarchical decomposition:
BAD by Foerster 2019 ICML - Bayesian Action Decoder for Deep Multi-Agent Reinforcement Learning

This recent AAMAS paper is based on peer evaluation and exchanging evaluation messages computed from recently obtained rewards:
PED-DQN by Hostallero 2020 AAMAS - Inducing Cooperation through Reward Reshaping based on Peer Evaluations in Deep Multi-Agent Reinforcement Learning
Some of the potential issues to discuss are: bandwidth usage of message exchange, message overhead in sharing the neighbors' rewards.

Using neighbors' information to achieve scalability in MARL most likely requires discussion of mean-field methods, such as:
Yang 2018 ICML - Mean Field Multi-Agent Reinforcement Learning.

- Going through the Appendices spurred a great deal of curiosity, as the authors mention that all agents share the same, synchronized random number generator with the same seed across all the agents. This leads me to believe that the philosophy of decentralized learning is lost in LToS. Synchronization is definitely not cost-free; all the more so if the synchronized RNG is used to sample an experience from the agents' replay buffers. How do the agents synchronize their RNG in a decentralized manner?

- In the Routing evaluation. has overhead been taken into account? How does LToS fare with respect to varied communications channel? What if the network were sparser? Do you observe any trends as you vary the extent of network connectivity?

---

> ### Author Response · Authors · 2020-11-22
> **Responses to Reviewer 5**
>
> > "it isn't obvious why and how relevant the cited works are... mentioning VDN, QMIX, and QTRAN (which together are some of the latest works in the factorization methods) does not seem to serve any further purpose, as they are no longer compared quantitatively or qualitatively to LToS... there appears to be no evidence whatsoever presented in the latter sections of the paper to show, let alone prove, the superior scalability of LToS."
>
> We position LToS at a particular line of research on networked MARL, where agents form a graph, have restricted communication (limited to neighboring agents), and cooperate on the objective of maximizing the average of cumulative rewards of all agents, following the setting of Zhang et al. (ICML’18), Qu et al. (NeurIPS’19), Chu et al. (ICLR’20), Qu et al. (NeurIPS’20). Actually, we do not argue that our method is very scalable while these factorization methods are totally not. We have revise the claims to make them precise and clear.
>
> Networked MARL focuses on decentralized learning as well as maximizing average return of all agents, while factorization methods are centralized training and focus the case all agents share a reward. Factorization methods can certainly be applied networked MARL. But in the literature it is empirically shown that QMIX performs poorly in large-scale networked MARL (Qu et al., 2020a). We also additionally performed the experiment of QMIX in 1) *traffic*, and it shows QMIX does not perform well as illustrated in Figure 4 and Table 2, and 2) *routing*, and it shows QMIX does not perform well either as illustrated in Figure 7 and Table 4.
>
> > Furthermore, some of the cited works have been left out at the evaluation stage, which leaves the reviewer puzzled as to which baselines LToS really hopes to outshine. The work needs some justification over why the following studies have not been compared to in the evaluation
>
> The main strength of LToS lies in "its capability to resolve selfishness and assign credits appropriately to bring about a harmonious cooperation in social dilemmas". LToS aims to bring a harmonious cooperation by reward sharing in networked MARL. Therefore, we compared LToS to the methods for networked MARL, such as ConseNet and NeurComm. Moreover, as the communication is limited in neighborhood, the methods of communication are not quite related. Additionally, DGN (Jiang et al. ICLR’20) is employed to properly handle the communication within neighborhood. That also makes us free from comparison with some methods of communication like CommNet (Sukhbaatar et al. NeurIPS 2016), because DGN already showed its advantage over CommNet by experiments when proposed.
>
> Eccles et al. (CoRR 2019) introduced two types of agents: innovator and imitator. There is an intrinsic reward added to the environment reward, so it is still one of the approaches that rely on hand-crafted reward designs, as summarized in Related Work. Moreover, the imitator needs to use the action of the innovator at each timestep to compute the intrinsic reward, which however is unrealistic in practice as they admitted in the paper.
>
> For BAD, all the experiments are performed in two-play cooperative games. It is non-trivial to extend it to more than two players. Besides, its hierarchical mechanism requires global information which is not realistic in our scenario.
>
> Hostallero et al. (AAMAS 2020) aim at maximizing social welfare, too. But unlike our work, they simply use temporal difference error for reward shaping instead of real reward sharing, and there is no explicit optimization for social welfare.
>
> Yang et al. (ICML 2018) just use mean-field method and neighbors' information to guarantee scalability and convergence to Nash equilibrium, but their goals don't contain global return optimization. Besides, DGN already showed its advantage over their MFQ by experiments when proposed.
>
> > Synchronization is definitely not cost-free; all the more so if the synchronized RNG is used to sample an experience from the agents' replay buffers. How do the agents synchronize their RNG in a decentralized manner?
>
> As agents cooperate on maximizing social welfare (not competitive setting), they can simply use a pre-defined RNG. Or, do we misunderstand your question?
>
> > In the Routing evaluation. has overhead been taken into account? How does LToS fare with respect to varied communications channel? What if the network were sparser? Do you observe any trends as you vary the extent of network connectivity?"
>
> Routing is a very complex problem and we only test LToS in a simplified scenario. We did not investigate varied communication channel of backbone network and network connectivity. More thorough investigation in routing will be considered in future work.

---

### Author Response · Authors · 2020-11-22
**To all the reviewers**

We thank all the reviewers for the insightful comments. For the main concerns regarding CTDE methods, we have revised our claims in the context of literature and performed additionally experiments of CTDE (i.e., QMIX) in traffic and routing. The results show QMIX does not perform well. Please refer to the revision for details.

---

### Decision · Program_Chairs · 2021-01-07
**Final Decision**

**Decision:**

Reject

**Comment:**

Although there was some initial disagreement on this paper, the majority of reviewers agree that this work is not ready for publication and can be improved in various manners. After the discussion phase there is also serious concern that the experiments need more work (statistically), to verify if they hold up. More comparisons with baselines are required as well. The paper could also be better put in context with the SOTA and related work. The paper does contain interesting ideas and the authors are encouraged to deepen the work and resubmit to another major ML venue.